# Proportion of foetal and placental implantation abnormalities in Madagascar: A cross-sectional study of 35,919 women at public-sector primary healthcare facilities in central and southern Madagascar, 2017–2020

Franziska Krätzig[1], Jie Mei[2,3], Mahery Rebaliha[4], Zavaniarivo Rampanjato[4,5,6,7], Rinja Ranaivoson[4], Jenia Razafinjato[4], Jan-Walter De Neve[5], Mara Anna Franke[6], Nadine Muller[4,5,6,8], Julius Valentin Emmrich[4,5,6,9,10]*

1 Charité - Universitätsmedizin Berlin, Institute of Tropical Medicine and International Health, Berlin, Germany, 2 Western Institute for Neuroscience, University of Western Ontario, London, Ontario, Canada, 3 Department of Computer Science, University of Western Ontario, London, Ontario, Canada, 4 Doctors for Madagascar, Antananarivo, Madagascar, 5 Heidelberg Institute of Global Health, Medical Faculty and University Hospital, University of Heidelberg, Heidelberg, Germany, 6 Global Digital Health Lab, Charité - Universitätsmedizin Berlin, Berlin, Germany, 7 Ministry of Public Health of the Republic of Madagascar, Antananarivo, Madagascar, 8 Department of Infectious Diseases and Respiratory Medicine, Charité - Universitätsmedizin Berlin, Berlin, Germany, 9 Charité Global Health and Department of Experimental Neurology and Center for Stroke Research, Charité - Universitätsmedizin Berlin, Berlin, Germany, 10 Berlin Institute of Health, Berlin, Germany

☯ These authors contributed equally to this work.
* julius.emmrich@charite.de

## Abstract

### Background

Like other countries in sub-Saharan Africa, Madagascar has a high burden of maternal and neonatal mortality. However, as the proportion of foetal and placental abnormalities among the Malagasy population is unknown, strategies aimed at reducing maternal and neonatal mortality are challenging to define and implement.

### Methods

We conducted a multi-year, cross-sectional study using secondary NGO data on obstetric ultrasound, including patient records of all pregnant women who received an obstetric ultrasound screening between July 1st, 2017, and September 30th, 2020, at 62 public-sector primary care facilities in urban and rural regions of Madagascar. We analysed demographic characteristics and determined the prevalence of foetal and placental abnormalities.

### Results

The dataset included 38,688 ultrasound screening reports from 35,919 women, where 2,587/35,919 (7.20%) women had more than one ultrasound exam. Most women (68.63%, 26,550/38,688) received their first ultrasound during the third trimester of pregnancy. Foetal

**Data Availability Statement:** The dataset supporting the findings of this study is openly available in Figshare, a public data repository, at 10.6084/m9.figshare.26494612.

**Funding:** The author(s) received no specific funding for this work.

**Competing interests:** The authors have declared that no competing interests exist.

malpresentation at 36 weeks of gestation or later was diagnosed in 5.48% (176/3,211) of women with the breech presentation being most common (breech 3.99%, 128/3,211; transverse 0.84%, 27/3,211; mobile 0.5%, 16/3,211; oblique 0.16%, 5/3,211). Placenta previa was found in 2.31% (875/38,755) and multiple gestations in 1.03% (370/35,919) cases. Around one in every 150 women (0.66%, 234/38,702) had amniotic fluid disorders.

## Conclusion

The proportion of foetal and placental abnormalities detected by obstetric ultrasound is consistent with findings from other countries in sub-Saharan Africa. In contrast to current WHO recommendations, pregnant women, particularly those from rural, resource-constrained settings attend obstetric ultrasound screenings most commonly during their third trimester of pregnancy.

## Background

Obstetric ultrasound is a non-invasive, sensitive, accurate and cost-effective imaging technique for the detection and diagnosis of congenital abnormalities, obstetric pathologies, and complications of pregnancy. The use of obstetric ultrasound is proliferating in resource-constrained settings due to its increased affordability, availability, portability, and durability [1]. Identifying foetal and placental abnormalities during pregnancy is essential to identify potential complications during gestation or delivery [2]. Foetal abnormalities, e.g., congenital malformations can necessitate additional provisions for delivery care, for example by requiring a Caesarean Section [3]. Equally, placental abnormalities such as placenta praevia or placenta accreta may require additional precautions during delivery [2]. Placental abnormalities may affect the development of the foetus in utero, requiring additional antenatal or postnatal care and supervision [4]. As such, accurate information on placental and foetal abnormalities that can be obtained from obstetric ultrasounds is vital for adequate antenatal support, delivery planning and reducing maternal and neonatal mortality. In resource-constrained settings, the WHO recommends obstetric ultrasound screenings once during pregnancy, before 24 weeks of gestation [5, 6]. For resource-constrained settings, the WHO questions the overall medical value of obstetric ultrasound but simultaneously promotes its ability to encourage pregnant women to attend routine antenatal care [6]. Nevertheless, reported clinical practices in low- and middle-income countries (LMICs) differ from this recommendation: Late obstetric ultrasound is routinely used to identify foetal and placental abnormalities (such as foetal malpresentation after 36 weeks of gestation or placenta previa) to reduce adverse events during delivery [7–14]. However, as these reports do not represent quasi-experimental or randomised trials, the effect of performing late instead of early obstetric ultrasound in resource-constrained settings to improve patient management and health outcomes remains elusive [1, 9, 11, 12, 15–19], especially given the challenges of administering obstetric ultrasound in resource-constrained settings, e.g., shortage of qualified staff and guidelines, proper supervision, and adequate maintenance of facilities and equipment [20].

In Madagascar, maternal and neonatal mortality rates remain high, with 353 maternal deaths per 100,000 live births in 2017 and a neonatal mortality of 20.6 per 1,000 live births in 2018 [21, 22]. Only approximately half of pregnant women in Madagascar complete 4 antenatal care visits as recommended by the WHO, while 54% of deliveries take place without

qualified personnel [23]. Eighty-three percent of Malagasy women report difficulties accessing healthcare services and women living in rural areas are particularly affected [24]. Common barriers to maternal and neonatal care include geographical distance to a healthcare facility, absence of a reliable and efficient mode of transportation, financial situations, cultural beliefs, a lack of family support, and poor quality of care [25–28]. Moreover, to the best of our knowledge, public databases focusing on the proportion of foetal and placental abnormalities are not available for the Malagasy population, which poses a challenge for policymaking and implementations regarding priority setting and design for programs aimed at improving maternal health in Madagascar.

To complement routine antenatal care in line with current WHO recommendations, the non-governmental organisation (NGO) Doctors for Madagascar has made mobile obstetric ultrasound screenings available to all pregnant women at 62 public-sector healthcare facilities in urban and rural regions of Madagascar, leading to a multi-year initiative to promote maternal healthcare and facilitate a data-driven understanding of foetal and placental abnormalities.

The aim of this study was to describe the demographic characteristics of a large group of pregnant women who utilized obstetric ultrasound services provided through an NGO. Specifically, we aimed to determine the prevalence of foetal and placental abnormalities in this study sample and to assess the timing and frequency of ultrasound examinations among these women. We expect these data to guide policymakers and program implementers in refining the design, implementation, and resource allocation of maternal health initiatives in this setting.

## Methods

### Study setting

This study was conducted in the Analamanga region (3.6 million inhabitants) in southern Madagascar, including Antananarivo, the capital of the island as well as the Atsimo-Andrefana region (1.8 million inhabitants), Anosy region (833,000 inhabitants), and Androy region (903,000 inhabitants) [29] (Fig 1). In Analamanga, 65% of women attend at least 4 antenatal care visits and two-thirds (68%) of deliveries take place in a healthcare facility [30]. Atsimo-Andrefana, Anosy, and Androy are among the most remote regions of the island [29]. In these regions, only around half of all women attend at least 4 antenatal care visits (50%, 49%, and 43% for Atsimo-Andrefana, Anosy, and Androy, respectively) and only approximately one-third of deliveries take place in a healthcare facility (30%, 49%, and 26% for Atsimo-Andrefana, Anosy, and Androy, respectively) [30]. Over 90% of the population in the three regions live below the international poverty line of 1.90 USD per day [28].

### Study design

The study is a cross-sectional study of secondary program data obtained from the NGO Doctors for Madagascar (DFM). Since 2017, DFM conducted mobile obstetric ultrasound screenings available to all pregnant women at public-sector healthcare facilities in the Analamanga region and the three southern regions Atsimo-Andrefana, Anosy and Androy.

### Obstetric ultrasound screening

Obstetric ultrasound exams were performed in mobile clinics by five midwives who obtained a diploma in obstetric ultrasound from the Malagasy Institute of Public and Community Health before the start of the program. The ultrasound teams consisted of a midwife who performed the ultrasound and one assistant, who were dispatched to each participating public healthcare

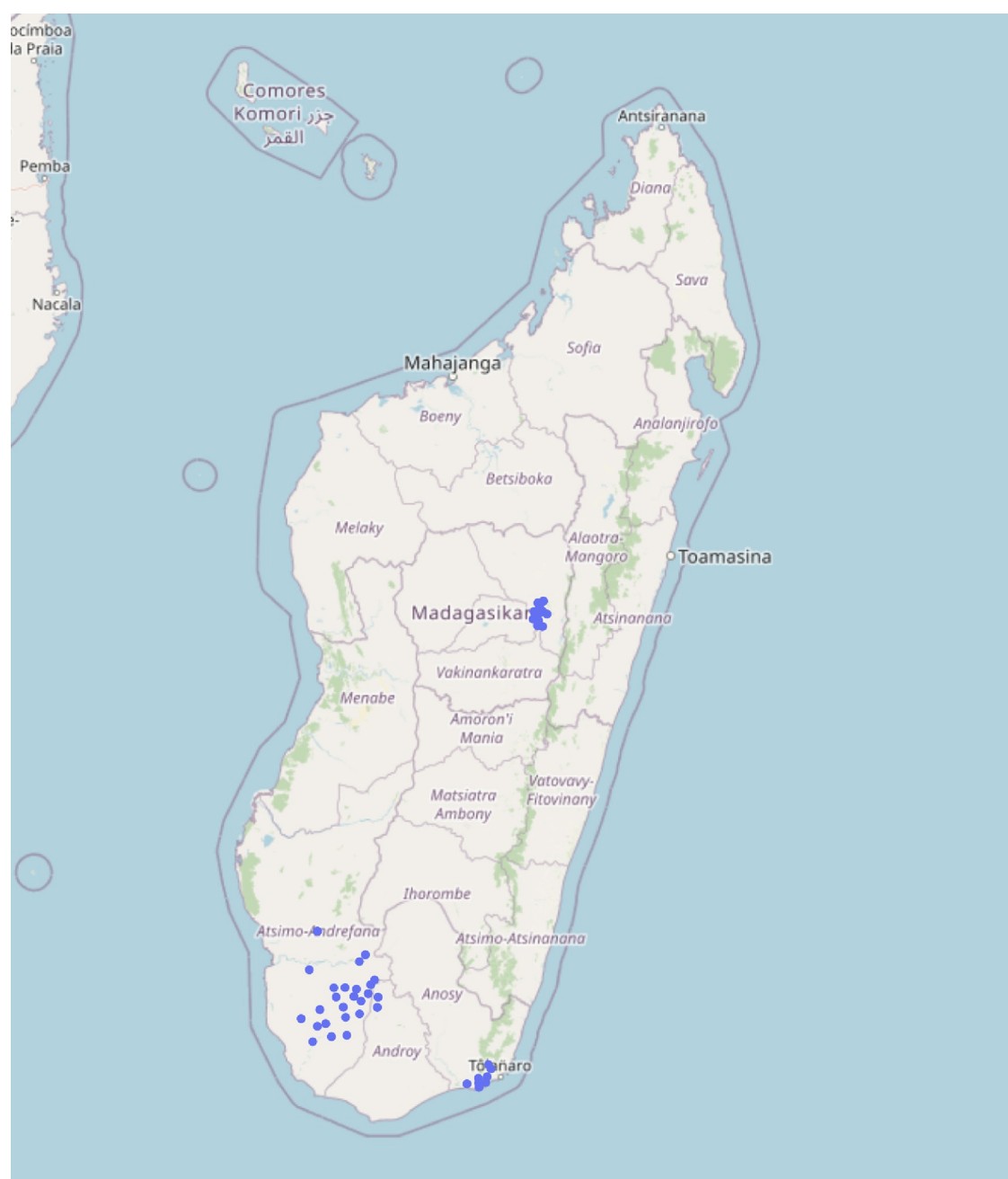

**Fig 1. Map of study facilities.** Map of Madagascar, depicting sixty-two public-sector primary care facilities where obstetric ultrasound screenings were conducted between July 1, 2017, and September 30, 2020, in urban (Analamanga) and rural (Atsimo-Andrefana, Anosy, and Androy) settings. The base map was provided by OpenStreetMap. OpenStreetMap is open data and is made available under the CC BY-SA license (OpenStreetMap contributors, http://www.openstreetmap.org/copyright) [31].

facility at least once per month complementing routine antenatal care activities. Mobile clinics were equipped with battery-powered mobile ultrasound devices (EDAN DUS60, Shenzhen, China). Pregnant women attending routine antenatal care at the facility could be referred for an obstetric ultrasound exam by the healthcare worker. All pregnant women were eligible to receive an obstetric ultrasound and there was no limitation based on gestational age.

Ultrasound was performed free of charge. In case the exam revealed a potentially adverse pregnancy outcome, women were either referred to an obstetric referral hospital or followed up by community- and facility-based healthcare workers.

Obstetric ultrasound screening was rolled out sequentially starting in July 2017 (Atsimo-Andrefana), October 2017 (Anosy), November 2017 (Androy), and August 2018 (Analamanga). In total, 30 (Analamanga), 18 (Atsimo-Andrefana), 11 (Anosy), and 3 (Androy) public-sector primary care facilities were included. Healthcare facilities in Analamanga were classified as urban; healthcare facilities in Atsimo-Andrefana, Anosy, and Androy were classified as rural. Healthcare facilities were chosen to participate in the obstetric ultrasound screening program upon agreement with senior officials from the Malagasy Ministry of Health.

## Definition of ultrasound findings

**Foetal malpresentation.** Foetal malpresentation was defined as a foetus engaging with the lower pole of the uterus in a position other than cephalic. Malpresentation included breech, oblique and transverse lie of the foetus [32–35].

**Location of the placenta.** Placenta previa was diagnosed when the placenta lay low in the uterus at $\geq$ 20 weeks gestation [36, 37]. Placenta previa was classified into the four following grades:

- *Grade I (minor previa)*: lower edge of placenta inside the lower uterine segment,

- *Grade II (marginal previa)*: The lower edge of the placenta reaches the internal opening of the cervix,

- *Grade III (partial previa)*: placenta partially covers the cervix, and

- *Grade IV (complete previa)*: The placenta completely covers the cervix.

**Amniotic fluid disorder.** The amniotic fluid volume was determined using the amniotic fluid index. To calculate the amniotic fluid index, the anteroposterior diameters of the largest empty fluid pocket in each quadrant of the amniotic sac were added together. An amniotic fluid index between 7 and 25 cm, with each pocket of fluid ranging from 2 to 8 cm, was considered normal. Fluctuations outside this range were classified as oligohydramnios (too little amniotic fluid) or polyhydramnios (too much amniotic fluid) [38, 39].

**Foetal malformation.** If a woman accessed the obstetric ultrasounds in the second trimester, the foetus was screened for structural abnormalities, including those of the head, spine, face, heart, abdominal wall, and skeleton.

**Gestational age.** Gestational age was either calculated using obstetric ultrasound measurements or by determining the number of days since the beginning of the last menstrual period. During the first trimester, gestational age was calculated from crown-rump length [40]; during the second and third trimester, gestational age was calculated from head circumference, abdominal circumference and femur length [41].

## Data analysis

**Data cleaning.** After conducting ultrasound exams, we obtained and anonymised ultrasound reports and performed a descriptive analysis of data from all pregnant women who underwent an obstetric ultrasound screening between July 1, 2017, and September 30, 2020.

Upon completion of the ultrasound scan, the examiner entered the scan results into an Excel database, leading to digitised reports for all women who had an obstetric ultrasound during the study period. All entries were anonymised to protect participants' personally

identifiable information. Data extracted from ultrasound exam reports included demographic information (age, village of residence, name of healthcare facility where ultrasound was performed), as well as maternal (location of the placenta, amniotic fluid volume, gestational age), and foetal (presentation of the foetus, number of foetuses, foetal malformation) characteristics. Data were first accessed for research purposes on 04/01/2021.

Data were entered into a standardised form, crosschecked, and screened for duplicates, out-of-range values, data entry errors, and overall consistency by one investigator (JM). Following each data-cleaning step, quality checks were conducted and the whole dataset was randomly sampled and screened by two investigators (FK and JM) for validity and consistency. For categorical data (i.e., foetal malformation), two clinicians (FK and JE) assigned numerical values to findings described in free texts to create categorical variables. The distance (in km) between the healthcare facility where the ultrasound was performed and a woman's village of residence was obtained from local NGO staff familiar with the area. Distances were calculated using the measure distance command in Google Maps. Assistance was also sought from NGO staff to resolve questions and inconsistencies in the data, which could not be solved by the investigators. Data were then processed using NumPy version 1.17.4 and pandas 0.25.3 [42, 43].

**Detection of multiple visits to healthcare facilities by the same individual.** To assess the dataset for more than one ultrasound exam conducted on the same individual over time in the absence of a unique personal identifier, we considered the following characteristics to be indicative of repeated ultrasound exams on the same individual: i) Similarity rating between names, as indicated by the Levenshtein distance (i.e., a metric that measures the difference between two sequences, calculated through the minimum number of single-character edits, such as deletions, insertions and substitutions required to change one string into another string [44]), of greater than > 50%; ii) matching age; iii) match between healthcare facility where ultrasound was performed and a woman's village of residence, iv) a shorter-than-240-day time interval between exams. Data entries missing any of the above information (429/38,688, 1.11%) were not included in detecting multiple visits. All detected revisits were screened by two investigators (JM and JF) to exclude false positives (n = 1,101) and entries with missing data (n = 86). For automatic detection of repeated exams, NumPy version 1.17.4 and pandas 0.25.3 were used [38, 39]. Investigators had no access to participants' personally identifiable information at any stage of the detection process.

## Study cohort

Ultrasound reports from all women who had at least one obstetric ultrasound examination at any of the sixty-two public-sector primary care facilities during the study period were included. A total of 38,688 ultrasound exam reports (data entries) were collected, from which we identified 33,332 individuals who had one single ultrasound examination and 2,587 individuals who had more than one ultrasound examination. Accordingly, an individual may have had multiple data entries depending on the number of ultrasound examinations she had. Therefore, the 38,688 data entries belonged to 35,919 women. Of these 38,688 reports, 492 lacked data on the time of the first ultrasound, and 827 lacked data on placenta location. Of these 35,919 women, 762 had missing information on the number of foetuses. Of all foetuses examined, information on amniotic fluid volume was missing in 1,012 cases. Fig 2 below illustrates the data source and study sample in more detail.

## Analysis

Our analysis proceeded in three steps. First, we assessed the socio-demographic characteristics of the whole study sample. We also determined the distance between the village of each study

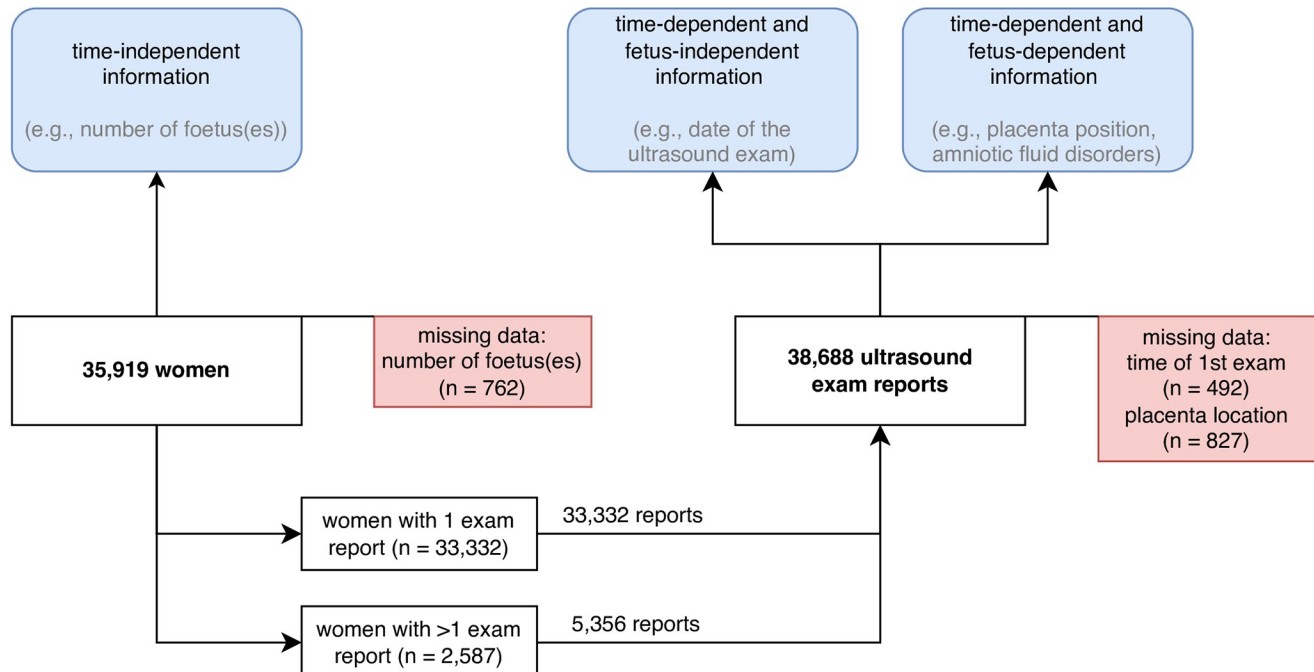

**Fig 2. Study cohort.** Depiction of study sample of women included in the study between July 1, 2017, and September 30, 2020, in urban (Analamanga) and rural (Atsimo-Andrefana, Anosy, and Androy) Madagascar.

participant and the primary healthcare centre. Visualisations were performed using Seaborn (version 0.11.0), a Python data visualisation library based on Matplotlib [45, 46]. The Shapiro-Wilk test was used for testing normality. For data that followed a normal distribution, we reported the sample mean and 95% confidence interval. For data that violated the normality assumption, the median and interquartile range (IQR) were reported. Secondly, we estimated the proportion of four groups of foetal and placental abnormalities, defined as (i) abnormal placenta position, (ii) number of foetuses above one, (iii) amniotic fluid disorders, and (iv) other placental and foetal pathologies respectively (e.g., placental adherence disorders, foetal malformations, and death).

For all parameters containing results that do not change over time in the same individual (e.g., number of foetuses), we used the number of individuals we have identified from the 38,688 ultrasound exam reports (n = 35,919, the sum of 33,332 individuals who had one ultrasound examination and 2,587 individuals who had more than one ultrasound examination) for descriptive statistics. For all parameters containing results that can change over time in the same individual (time-dependent but foetus-independent, e.g., timing of the ultrasound exam) we used the number of all ultrasound examination records/data entries (n = 38,688, the total number of ultrasound examination records obtained from 35,919 individuals, where one individual may have multiple data entries due to twin pregnancy or follow-up exams) for descriptive statistics and then inspected data for multiple observations per person over time. For all parameters containing results that can change over time and differ across foetuses in the same individual (time- and foetus-dependent, e.g., abnormal placenta position, amniotic fluid disorders), we used the number of all ultrasound examination records (n = 38,688), including additional information on 63 foetuses from multiple gestations of abnormal placenta positions (n = 38,755) and additional information on 14 foetuses from multiple gestations of amniotic

fluid disorders (n = 38,702). Thirdly, we conducted Pearson's chi-squared test to test for differences in characteristics for women from urban and rural backgrounds. We equally used Pearson's chi-squared test (for categorical data) and Spearman rank-order correlation coefficient (for continuous data) to analyse correlations between foetal and placental abnormalities and the number of ultrasound examinations. A p-value below 0.05 was considered statistically significant.

### Data availability and ethics

For this study, we reached a data use agreement with the NGO Doctors for Madagascar. Data is available upon request from Doctors for Madagascar (info@doctorsformadagascar.com). This study has been approved by the Ethics Committee of the University of Heidelberg (registration number S-854/2020). We obtained formal approval to conduct this secondary analysis of de-identified NGO data from the district health office, a regional sub-division of the Malagasy Ministry of Health, of Atsimo-Andrefana. All methods were conducted following relevant guidelines and regulations. Informed consent was waived by the Ethics Committee of the University of Heidelberg due to the retrospective nature of the study.

## Results

### Sample description

Our analytical sample included 35,919 women with 38,688 ultrasound reports. Data on age was available from 84.04% (30,186/35,919) of women; the median age was 22 years (IQR: 18–28) (Fig 3). We found age heaping at ages that are multiples of ten, which is common in low-resource settings. Most women (84.96%; 30,517/35,919) lived in rural areas; 15.04% (5,402/35,919) of women lived in urban areas.

### Distance between the healthcare facility and the village of residence

The median distance between the ultrasound-providing healthcare facility and a woman's village of residence was 6 km (range: 0.7–14 km). For rural regions the median distance was 8 km (range: 1–16 km); for urban regions, the median distance was 1.9 km (range: 0.5–3.2 km, Fig 4).

### Timing of ultrasound exams during pregnancy

Among all women, 7.20% (2,587/35,919) had more than one ultrasound examination. Of those, 6.78% (2,434/35,919) received two ultrasound examinations, and only a few received three or four ultrasound exams during their pregnancy (0.41% (149/35,919) and 0.01%, (4/35,919), respectively). Most visits for obstetric ultrasound to the mobile clinic occurred during the third trimester of pregnancy (68.63%; 26,550/38,688), while 25.57% (9892/38,688) and 4.53% (1754/38,688) of obstetric ultrasound examinations occurred during the second and first trimester of pregnancy, respectively. Most scans were performed between 26 and 37 weeks of gestation (Table 1). The proportion that received a scan in the third trimester was significantly higher among women from rural backgrounds (71.33%, 23,352/32,740) than from urban backgrounds (53.77%, 3,198/5,948; p-value: <0.01).

For the 2,587 women who had more than one ultrasound examination, we also analysed when the follow-up visits occurred. Among these 2,587 women, inconsistencies between date of facility visit and trimester (e.g., earlier date is associated with a later trimester) or missing information were observed in data from 19 women, leading to exclusion from this analysis. Of the 2,568 women included, 2,418 had two ultrasound examinations (1 revisit; 94.16%), 147

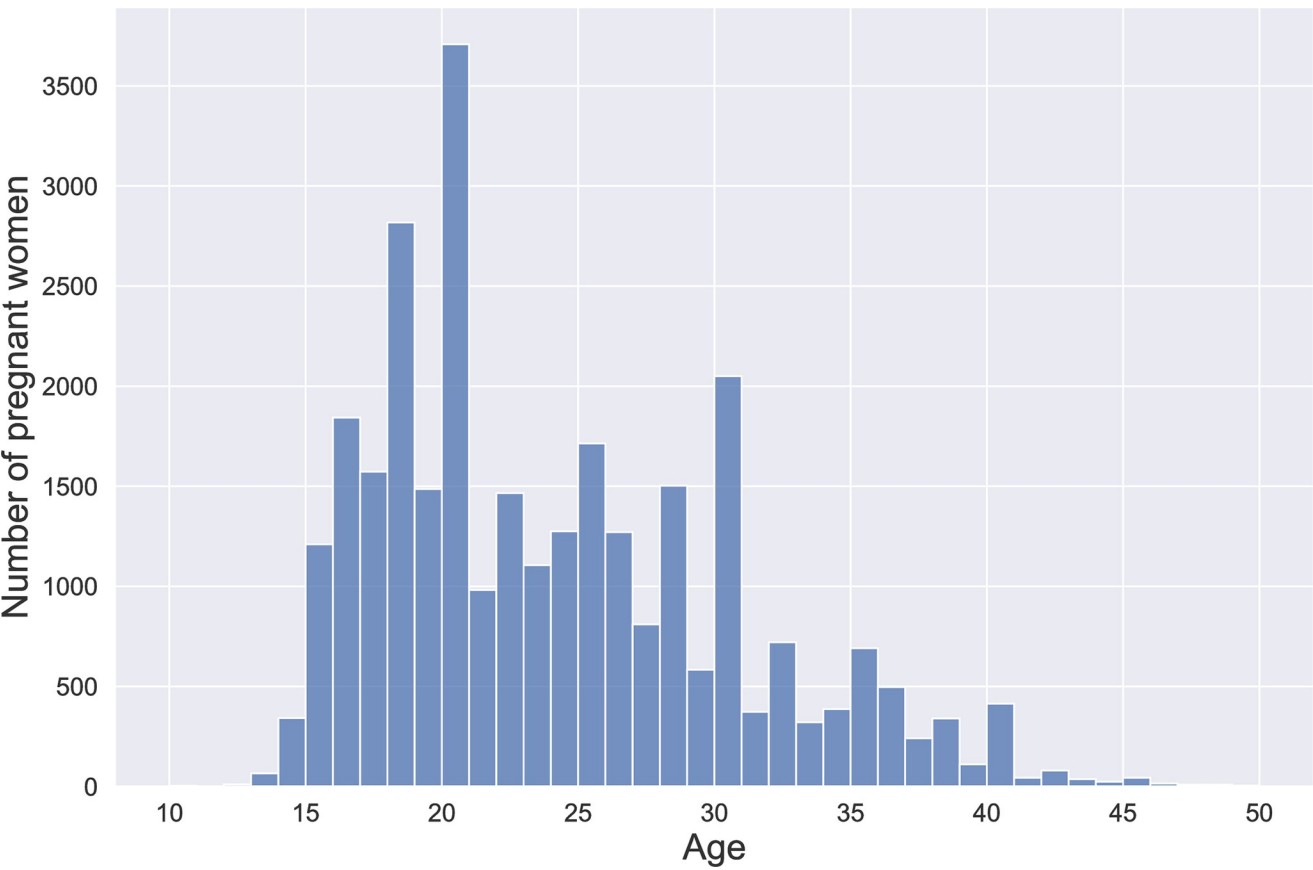

**Fig 3. Age distribution of study participants.** Age distribution of 35,919 pregnant women who received an obstetric ultrasound screening exam between July 1, 2017, and September 30, 2020.

had three ultrasound examinations (2 revisits; 5.72%), and 3 had four ultrasound examinations (3 revisits; 0.12%).

Among the 2,418 women who had two ultrasound examinations, a total of 256, 1433, and 729 first examinations occurred in the first, second, and third trimesters, respectively, and the follow-up occurred in the first trimester in 6 women, in the second trimester in 150 women, and in the third trimester in 2262 women. Of the 147 women who had three ultrasound examinations, 35 had their first examination in the first trimester, 95 had their first examination in the second trimester, and 17 had their first examination in the third trimester. The first follow-up was in the second trimester in 48 women and in the third trimester in 99 women, respectively. A total of 2 women had the second follow-up in the second trimester, and 145 women in the third trimester. Of the 3 women who had four ultrasound examinations, 2 had their first examination in the first trimester and 1 in the second trimester. The first follow-up occurred was in the second trimester for one woman, and in the third trimester for 2 women. The second and third follow-ups were all in the third trimester.

## Proportion of foetal and placental abnormalities

Data on foetal presentation between 24 and 43 weeks of gestation was available for 99.62% of all (29,674/29,788) women and missing for 0.38% (114/29,788). From $\geq$ 36 weeks of gestation, 94.52% (3,035/3,211) of all foetuses were in a cephalic position. Of the remaining foetuses

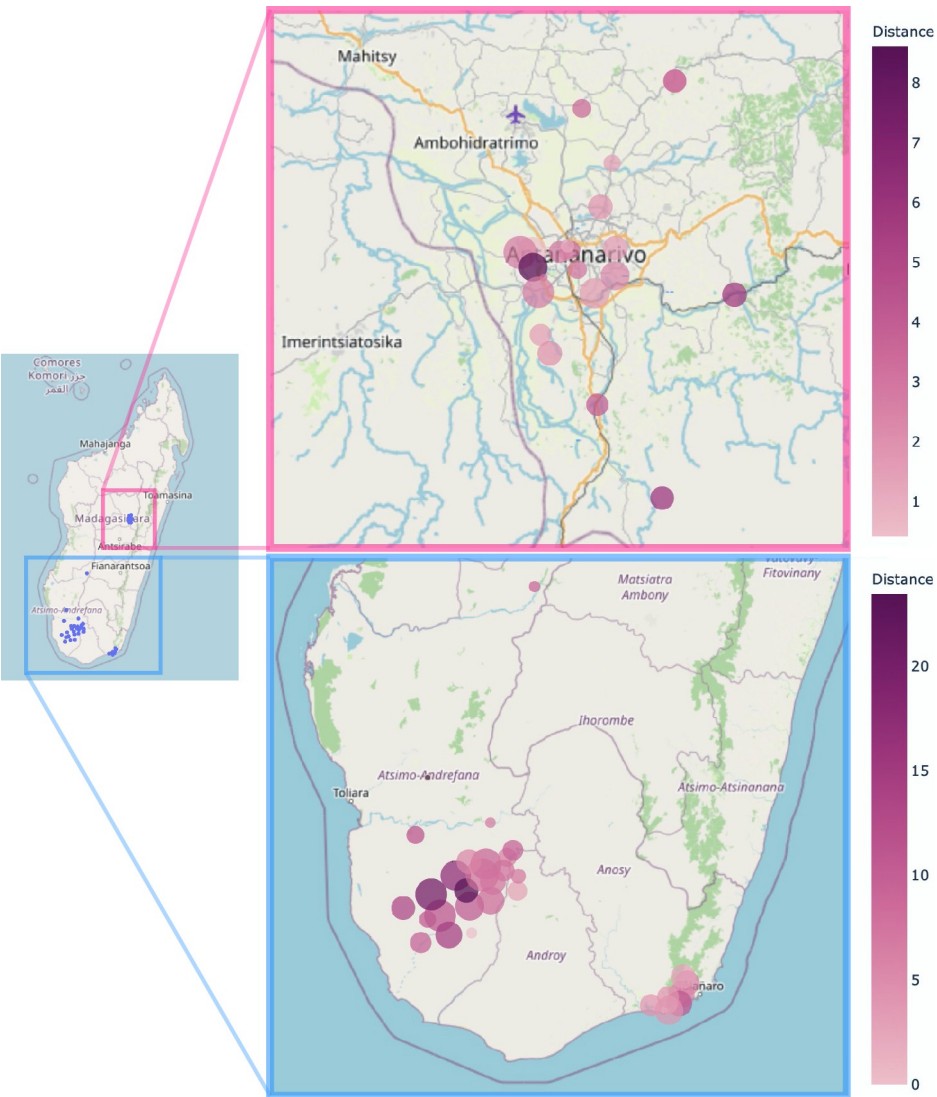

**Fig 4. Distance between the nearest healthcare facility and a woman's village of residence.** Average distance from a woman's village of residence to the nearest public-sector primary care facility where obstetric ultrasound screenings were performed for a) urban, and b) rural regions of Madagascar. size of bubbles = number of women that visited a healthcare facility; colour intensity of bubbles = average distance from each woman's village of residence to the nearest public-sector primary care facility where obstetric ultrasound screenings were performed. The base map was provided by OpenStreetMap. OpenStreetMap is open data and is made available under the CC BY-SA license (OpenStreetMap contributors, http://www.openstreetmap.org/copyright) [31].

from ≥ 36 weeks of gestation, 3.99% (128/3,211) were in the breech, 0.84% (27/3,211) were in transverse, 0.5% (16/3,211) were in mobile and 0.16% (5/3,211) were in an oblique position. The prevalence of breech presentation decreased with gestational age. Around one in six foetuses (15.67%, 824/5,258), 9.5% (972/10,236), 4.39% (481/10,969), and 3.99% (128/3,211) were in breech position between 24 to 27, 28 to 31, 32 to 35, and from 36 weeks of gestation, respectively.

Placenta previa was observed in 2.31% (875/38,755) cases. Of those, 76,11% (666/875), 7.66% (67/875) and 0.69% (6/875) were placenta previa grade I, II, and III, respectively; 15.54% (136/875) diagnosed with placenta previa where not further categorised (Table 2). We

**Table 1. Demographics and maternal characteristics.**

|  | Category | % | n |
|---|---|---|---|
| **Region (N = 35,919)** | Rural | 84.96 | 30,517 |
|  | Urban | 15.04 | 5,402 |
| **Repeated ultrasound exams (N = 2,587)** | Rural | 80.83 | 2,091 |
|  | Urban | 19.17 | 496 |
|  | Missing | 0 | 0 |
| **Timing of first obstetric ultrasound (N = 38,688)** | 1st trimester* | 4.53 | 1,754 |
|  | 2nd trimester' | 25.57 | 9,892 |
|  | 3rd trimester" | 68.63 | 26,550 |
|  | Missing | 1.27 | 492 |
| **Number of ultrasound exams (N = 35,919)** | 1 | 92.80 | 33,332 |
|  | 2 | 6.78 | 2,434 |
|  | 3 | 0.41 | 149 |
|  | 4 | 0.01 | 4 |

*1–12 weeks of gestation; '13–24 weeks of gestation;"25–40 weeks of gestation.

found that 96.85% (34,787/35,919) of women had one foetus and 1.03% (370/35,919) had multiple gestations. Overall, 1.02% (366/35,919) had a twin pregnancy and 0.01% (4/35,919) had three foetuses. Amniotic fluid disorders were diagnosed in 0.66% (234/38,702) of cases. Low volume of amniotic fluid (oligohydramnios) was diagnosed in 12.82% (30/234) of cases and high volume of amniotic fluid (polyhydramnios) was seen in 87.18% (204/234) (Table 2). In terms of foetal pathologies, non-gravid uteruses (0.33%; 118/35,919), foetal death in utero

**Table 2. Foetal and placental abnormalities.**

|  | Category | % | n |
|---|---|---|---|
| **Abnormal placenta position (N = 38,755)** | Yes | 2.31 | 875 |
|  | Placenta previa I | 76.11 | 666 |
|  | Placenta previa II | 7.66 | 67 |
|  | Placenta previa III | 0.69 | 6 |
|  | Placenta previa, no categorisation | 15.54 | 136 |
|  | No | 97.69 | 37,053 |
|  | Missing | 2.13 | 827 |
| **Number of foetuses (N = 35,919)** | 1 | 96.85 | 34,787 |
|  | 2 | 1.02 | 366 |
|  | 3 | 0.01 | 4 |
|  | Multiple gestations | 1.03 | 370 |
|  | Missing | 2.12 | 762 |
| **Amniotic fluid disorders (N = 38,702)** | No | 96.45 | 37,456 |
|  | Polyhydramnios | 87.18 | 204 |
|  | Oligohydramnios | 12.82 | 30 |
|  | Missing | 2.61 | 1,012 |
| **Placental and foetal pathologies (N = 35,919)** | Non-gravid uterus | 0.33 | 118 |
|  | Molar pregnancy | 0.03 | 12 |
|  | Extrauterine pregnancy | 0.03 | 11 |
|  | Malformation | 0.06 | 22 |
|  | Foetal death in utero | 0.19 | 68 |

(0.19%; 68/35,919), and foetal malformation (0.06%; 22/35,919) including acrania or hydrocephalus were the most common pathologies (Table 2).

Additional analyses showed no significant correlation between the number of foetuses, amniotic fluid disorders, or placental and foetal pathologies and the number of ultrasound visits. Pearson's chi-square test revealed a significant correlation between the grade of placenta praevia and the number of revisits (p <0.01). We have uploaded the tables depicting the results of these analyses in S1 File.

## Discussion

Using data collected from over 35,000 women, we estimated the proportion of foetal and placental abnormalities among pregnant women attending obstetric ultrasounds at public-sector healthcare facilities in Madagascar. Our study revealed three main findings: First, most women in our sample came from rural areas. Secondly, less than 10% of women in our sample were diagnosed with a potential complication during pregnancy. Foetal malposition, breech position and abnormal position of the placenta were the most common findings. Lastly, most obstetric ultrasounds were conducted in the third trimester of pregnancy, especially among women from rural backgrounds.

About one-fourth of the Malagasy population live more than 10km from a healthcare facility [47]. This holds especially true for women in rural areas where most of the Malagasy population and individuals in this study reside [47]. We found that women from rural areas had to travel longer distances (median 8km) than women from urban areas (median 1.9km) to reach a healthcare facility providing obstetric ultrasound. No other study has so far investigated the distance to the nearest available obstetric ultrasound in Madagascar; a systematic review that assessed the access to skilled care for childbirth in Sub-Saharan Africa found the distance to the nearest healthcare facility to be 4km and 15km in urban and rural settings, respectively [48]. Given that only around two-thirds of Madagascar´s inhabitants have access to any form of primary health care and that the longer the distance to a healthcare facility, the lower the access to maternal care [49, 50], women who live in villages further from primary healthcare facilities, especially those from rural backgrounds in our sample had to travel to access obstetric ultrasounds are in a challenging situation.

Overall, the proportion of foetal malpresentation and multiple gestations detected by obstetric ultrasound in our study are consistent with findings from other countries in sub-Saharan Africa [35, 51, 52], with a few important exceptions. First, in our sample, 5.48% (176/3,211) of all foetuses were in malpresentation from >36 weeks of gestation. These findings differ from a study from Uganda where foetal malpresentation in the third trimester was found in 15% of pregnancies, but are consistent with a study from Guatemala where the overall rate of foetal malpresentation was 4.49% and the prevalence of breech presentation was 3–4% of all term pregnancies [7, 35, 51]. This might be a consequence of different sampling strategies of the studies, given that the sample in Uganda only included women accessing a rural, private clinic, while our study also included public facilities. As, overall, care at public facilities is cheaper and more accessible in Sub-Saharan Africa and socioeconomic status is a key determining factor in accessing healthcare [27] it might be that women who fear complications are more likely to access healthcare at a private facility.

Secondly, while the overall prevalence of placenta previa in Sub-Saharan Africa is unclear and in a reported range between 0.27% and 1.7% [10, 17, 53], these results do not match the findings in our study where the prevalence of placenta previa was 2.31% (875/38,755). This discrepancy might be explained by the gestational age at the time of ultrasound. Around one-

fourth of ultrasounds in our study were performed during the second trimester when placental abnormalities were more common.

Thirdly, accurate data on the prevalence of congenital foetal anomalies is rare. Data from registries of congenital anomalies indicate that they are seen in 2–3% of newborns in Africa [54]. However, only 0.06% (22/35,919) of foetuses had a malformation detected during obstetric ultrasound, which is consistent with findings from other studies on the prevalence of foetal abnormalities during routine third-trimester ultrasound [55]. This might be an indication of the limited diagnostic capacities of obstetric ultrasounds in identifying neonatal abnormalities, especially given their dependence on the training and experience of the sonographer [7, 56, 57].

Lastly, the incidence of extrauterine pregnancy worldwide ranges from 1.02% to 3.9%, which is not in line with our findings of 0.03% (11/35,919) [58]. As extrauterine pregnancy is the main cause of first-trimester maternal mortality, the discrepancy in results is most likely due to the low number of ultrasound examinations during the first trimester (4.53% 1754/38,688) in our study [59].

A central finding of our study is that women in rural and urban Madagascar accessed obstetric ultrasound predominantly during late pregnancy. Two-thirds (68.63%; 26,550/38,688) of women had their first ultrasound examination during the third trimester of pregnancy and a quarter of women (25.57%; 9,892/38,688) during the second trimester. Rural women were significantly more likely to have their first ultrasound during their third trimester compared to women from urban backgrounds. These results are consistent with findings from other resource-restricted settings that show a clear trend towards accessing antenatal care late in pregnancy, despite the WHO recommendation of conducting obstetric ultrasounds before 24 weeks gestation [12, 13, 52, 60]. Evidence from other Sub-Saharan African countries suggests that women accessing obstetric ultrasounds late in pregnancy is a common phenomenon.

Some studies suggest that screening for breech presentation at 36–37 weeks of gestation in resource-restricted settings might reduce the risk of perinatal complications and maternal mortality [54, 55, 61]. Early ultrasounds, on the other hand, are crucial for understanding if a pregnancy is intrauterine, and for identifying essential indicators for reducing early maternal mortality, such as the number of foetuses and their respective vitality [51]. Our findings highlight the need for a better knowledge of the reasons women, especially those from rural regions, only access ultrasound examinations late in pregnancy, despite the WHO recommendation and advantages of early ultrasound. It further calls into question if additional late ultrasounds might be an important diagnostic tool in reducing perinatal mortality, even though the current evidence on this subject is limited [51].

Additionally, our study shows that there was no significant correlation between the number of ultrasounds visits and foetal and placental abnormalities that had been identified during obstetric ultrasounds, apart from the grade of placenta praevia. This is worrying as several of the pathologies identified should be followed up through repetitive ultrasound examinations in pregnancy. Our findings indicate that women in Madagascar with foetal and placental abnormalities do not access repetitive ultrasound examinations in such cases. Further indicate should be conducted to identify the reasons behind these healthcare seeking behaviours.

Our study also has limitations. First, ultrasound examinations were performed by different sonographers, which may have introduced inter-observer bias. However, all sonographers underwent a standardised training program and obtained a diploma in obstetric ultrasound from the National Institute of Public and Community Health to improve inter-observer reliability and reduce bias. Secondly, we based our analyses on primary data that was not collected for research purposes and contained data entry errors and missing data due to limited resources and under-standardised data acquisition guidelines. To enable optimised

preprocessing and cleaning, we designed and applied several methods, including crosschecking, screening for duplicate entries, out-of-range values, data entry errors, and overall consistency. In addition, as no unified personal identifier system was employed at the time of this study, it was difficult to detect multiple visits to the healthcare facilities by the same individual. Accordingly, we used several criteria to screen for multiple visits and conducted inspections of the whole dataset through random sampling of data entries, for improved validity and consistency. In addition, some, but not all women with twin pregnancies had two records/data entries per visit, one entry for each foetus. Meanwhile, in 20 women with twin pregnancies, the number of foetuses in the records was inconsistent. Nevertheless, it was not possible to correct these errors due to lack of reliable information given the time lag between data collection and analysis. Fourthly, as we drew on secondary data from an NGO program, our sample may not be representative of all women seeking antenatal care and results may differ for other populations and settings. Nevertheless, our dataset contains data from a variety of cultural, demographic, and geographic contexts within Madagascar, with substantial variation regarding their geography, health indicators, and socio-economic status. We compared the basic demographic characteristics of our sample with the broader population of women of reproductive age in Madagascar and found them to be comparable. Demographic characteristics of our study population were also similar to those in studies conducted in Nigeria and Uganda [10, 13]. We do acknowledge however, that this may not be an accurate comparison, as studies conducted in similarly resource-constrained settings in SSA may have been affect by similar limitations.

Key strengths of our study included its large sample size of rural and urban pregnant women from a resource-restricted setting. Furthermore, the secondary analysis of NGO program data allowed for the inclusion of marginalised or otherwise hard-to-reach communities with restricted access to health services, who are commonly underrepresented in studies and surveys within the fields of obstetrics and prenatal development and diagnosis. To our knowledge, this study represents the largest dataset with obstetric ultrasound data collected from Madagascar, featuring both urban and rural settings.

## Conclusion

Our data show that obstetric ultrasound in a resource-restricted setting can lead to the detection and diagnosis of high-risk obstetric conditions including placental abnormalities, malpresentation of the foetus, and multiple gestations, which are highly associated with maternal and foetal morbidity and mortality. In stark contrast to current WHO recommendations, our findings suggest that universally accessible obstetric ultrasound is commonly and mostly utilised by pregnant women, especially those from rural, resource-constrained settings, during the third trimester of pregnancy. This finding highlights the need to improve policy and practice in Madagascar to improve access to obstetric ultrasounds, particularly during the first and second trimesters.

## Supporting information

**S1 File.**
(DOCX)

## Acknowledgments

We would like to thank Doctors for Madagascar for initiating and running the ultrasound project, and all study participants in Madagascar for taking part in this project. We would equally like to thank Prof. Dr. Dr. Till Bärninghausen for his support of this study.

## Author Contributions

**Conceptualization:** Franziska Krätzig, Nadine Muller, Julius Valentin Emmrich.

**Data curation:** Franziska Krätzig, Jie Mei, Zavaniarivo Rampanjato, Rinja Ranaivoson.

**Formal analysis:** Franziska Krätzig, Jie Mei, Mahery Rebaliha, Zavaniarivo Rampanjato, Rinja Ranaivoson, Jenia Razafinjato, Julius Valentin Emmrich.

**Investigation:** Franziska Krätzig, Jie Mei, Mahery Rebaliha, Jan-Walter De Neve, Mara Anna Franke, Nadine Muller.

**Methodology:** Jie Mei, Jan-Walter De Neve, Nadine Muller, Julius Valentin Emmrich.

**Project administration:** Franziska Krätzig, Zavaniarivo Rampanjato, Rinja Ranaivoson, Jenia Razafinjato, Jan-Walter De Neve, Mara Anna Franke, Julius Valentin Emmrich.

**Resources:** Mahery Rebaliha, Zavaniarivo Rampanjato, Rinja Ranaivoson, Jenia Razafinjato, Mara Anna Franke.

**Supervision:** Jan-Walter De Neve, Nadine Muller, Julius Valentin Emmrich.

**Visualization:** Franziska Krätzig, Jie Mei.

**Writing – original draft:** Franziska Krätzig, Jie Mei.

**Writing – review & editing:** Franziska Krätzig, Jie Mei, Mahery Rebaliha, Zavaniarivo Rampanjato, Rinja Ranaivoson, Jenia Razafinjato, Jan-Walter De Neve, Mara Anna Franke, Nadine Muller, Julius Valentin Emmrich.

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
