## [Decision Letter · Decision Letter 0]

27 Mar 2024

PONE-D-24-05563

Proportion of foetal and placental abnormalities in Madagascar: a cross-sectional study of 35,919 women at public-sector primary healthcare facilities in central and southern Madagascar, 2017-2020

PLOS ONE

Dear Dr. Emmrich,

Thank you for submitting your manuscript to PLOS ONE. After careful consideration, we feel that it has merit but does not fully meet PLOS ONE’s publication criteria as it currently stands. Therefore, we invite you to submit a revised version of the manuscript that addresses the points raised during the review process.

We look forward to receiving your revised manuscript.

Kind regards,

Giovanni Tossetta, Ph.D

Academic Editor

PLOS ONE

Journal Requirements:

2. In this instance it seems there may be acceptable restrictions in place that prevent the public sharing of your minimal data. However, in line with our goal of ensuring long-term data availability to all interested researchers, PLOS’ Data Policy states that authors cannot be the sole named individuals responsible for ensuring data access (http://journals.plos.org/plosone/s/data-availability#loc-acceptable-data-sharing-methods).

3. We note that Figure 1 and 3 in your submission contain map/satellite images which may be copyrighted. All PLOS content is published under the Creative Commons Attribution License (CC BY 4.0), which means that the manuscript, images, and Supporting Information files will be freely available online, and any third party is permitted to access, download, copy, distribute, and use these materials in any way, even commercially, with proper attribution. For these reasons, we cannot publish previously copyrighted maps or satellite images created using proprietary data, such as Google software (Google Maps, Street View, and Earth). For more information, see our copyright guidelines: http://journals.plos.org/plosone/s/licenses-and-copyright.

a. You may seek permission from the original copyright holder of Figure 1 and 3 to publish the content specifically under the CC BY 4.0 license.  

Reviewers' comments:

Reviewer's Responses to Questions

**Comments to the Author**

1. Is the manuscript technically sound, and do the data support the conclusions?

Reviewer #1: Yes

Reviewer #2: Yes

2. Has the statistical analysis been performed appropriately and rigorously? 

Reviewer #1: Yes

Reviewer #2: I Don't Know

3. Have the authors made all data underlying the findings in their manuscript fully available?

Reviewer #1: Yes

Reviewer #2: Yes

4. Is the manuscript presented in an intelligible fashion and written in standard English?

Reviewer #1: Yes

Reviewer #2: Yes

5. Review Comments to the Author

Reviewer #1: Introduction

The introduction reads well but given that your title and focus of the paper if on foetal and placental anomalies, it would be good to have a paragraph on foetal anomalies as well some literature on placenta, particularly because the placenta is critical for foetal development.

It is also important to highlight, briefly the importance of obstetric ultrasound screenings and how cases with adverse events are reduced. There is some information on this in line 77 but a suggestion to expound briefly on this. This will broaden the readership of this interesting article.

Methods

Study setting

Kindly provide a reference for the statement from line 114 – 117 providing the statistics on the number of women who attend antenatal care visits within these remote regions.

Study design

Line 129 – 131 (Line beginning with: Following ultrasound exams…) can be moved to the Data Analysis section. This is because this refers to the reports where the data was obtained from.

Study sample

Line 222 - 229 – please consider summarizing this in a flow diagram. It will be much easier for the reader to understand the numbers.

Analysis

Line 241 - What was the criteria used to define placenta pathology?

Kindly clarify line 243 (Line starting with: For all parameters..) as this is not clear who exactly was included in the descriptive statistics.

The section is confusing as it is not clear who the 35919 and 38688 represent and why different participants were used in the analysis and there was no common denominator. Kindly provide some clarity.

Results

Was there a correlation between region (rural vs urban) and number of ultrasounds

For the women who had >1 ultrasound, which trimesters were they categorized under for their first and follow up visits?

Was the number of ultrasound exams associated with any of the factors in Table 2?

Table 1 if Missing is 0, it can be excluded from the table

Reviewer #2: The paper is overall well organized and well written with appropriate style and language.

The flow is logical and well structured mostly and the chosen style is consistent throughout. The tables presented are straightforward although graphs could have been used to make the paper more impressive.

What makes the paper interesting is that it covers a part of Sub-Saharan Africa, Madagascar which has a high burden of increased maternal and neonatal mortality. The study analyzed 38,685 ultrasound screening reports which is a very significant number along with the demographic characteristics.

However, the aims and objectives of the study are not well defined. They mention that strategies aimed at reducing maternal and neonatal mortality are challenging to define and implement. But a clear aim of the study is not clear. (Major Issue).

The limitations and strengths of the study have been discussed very clearly and that is worth mentioning.

With regards to the results obviously, most of the results cannot be accurately interpreted as the ultrasounds were not performed at the desired gestational ages and the fetal congenital abnormalities have been mentioned very briefly and with no clear description which is interpreted from the data.

The authors however should be credited for their job with the best of what could be expected from the data obtained from underdeveloped rural areas of Sub Saharan Africa.

To summarize I think this article is good and I do recommend it for publication after defining the objectives and mentioning suggestions in conclusion more distinctly.

6. PLOS authors have the option to publish the peer review history of their article (what does this mean?). If published, this will include your full peer review and any attached files.

Reviewer #1: No

Reviewer #2: No

---

## [Author Response · Author response to Decision Letter 0]

29 May 2024

Dear Editor,

First, we express our gratitude to the reviewers for their valuable and constructive input. We thank the reviewers for the positive feedback on our article and for recognising the value in presenting data from an understudied region. We are happy to hear that our paper is potentially acceptable for publication in your journal.

Enclosed is a detailed response addressing all reviewer and editor comments. We have incorporated the suggested changes, clearly indicated through track changes in the accompanying manuscript. 

We look forward to your feedback and thank you for your consideration.

Dr Julius Emmrich

---

## [Decision Letter · Decision Letter 1]

12 Jun 2024

PONE-D-24-05563R1Proportion of foetal and placental abnormalities in Madagascar: a cross-sectional study of 35,919 women at public-sector primary healthcare facilities in central and southern Madagascar, 2017-2020PLOS ONE

Dear Dr. Emmrich,

Thank you for submitting your manuscript to PLOS ONE. After careful consideration, we feel that it has merit but does not fully meet PLOS ONE’s publication criteria as it currently stands. Therefore, we invite you to submit a revised version of the manuscript that addresses the points raised during the review process.

We look forward to receiving your revised manuscript.

Kind regards,

Giovanni Tossetta, Ph.D

Academic Editor

PLOS ONE

Journal Requirements:

Reviewers' comments:

Reviewer's Responses to Questions

**Comments to the Author**

1. If the authors have adequately addressed your comments raised in a previous round of review and you feel that this manuscript is now acceptable for publication, you may indicate that here to bypass the “Comments to the Author” section, enter your conflict of interest statement in the “Confidential to Editor” section, and submit your "Accept" recommendation.

Reviewer #1: (No Response)

Reviewer #2: All comments have been addressed

2. Is the manuscript technically sound, and do the data support the conclusions?

Reviewer #1: Yes

Reviewer #2: Yes

3. Has the statistical analysis been performed appropriately and rigorously? 

Reviewer #1: Yes

Reviewer #2: I Don't Know

4. Have the authors made all data underlying the findings in their manuscript fully available?

Reviewer #1: Yes

Reviewer #2: Yes

5. Is the manuscript presented in an intelligible fashion and written in standard English?

Reviewer #1: Yes

Reviewer #2: Yes

6. Review Comments to the Author

**Reviewer #1: **Thank you for making the suggested edits on the revision now submitted.

Line 71 - 79: The paragraph included on foetal and placental abnormalities requires supporting references. Kindly include recent references to support the statements provided.

Line 115 - 121: Suggestion to use past tense, the aim of this study was to.. as you are now reporting on the findings

Line 141: Include a reference for Open Street Map as per the Journal guidelines

Line 248: Study sample can read study cohort

For the study sample section it will be easier to understand the numbers if depicted in a flow diagram. Kindly consider including one so that the readers can better understand the flow and final selection of study participants

Line 283: there is a bracket close but not sure where the opening bracket is

Line 302 and 303: The correlations can be depicted as graphs as this is briefly mentioned in Line 398 - 401. It is a important finding; and later discussed in line 477-480.

**Reviewer #2:** The manuscript has been revised and queries have been answered, but the limitation of the manuscript is quite big. The results drawn are not representing the exact fetal and placental pathology of the population due to the various reasons described in the limitations the most highlighted being the timing of ultrasound which is mainly 3rd trimester.

Similarity of results with other Sub Sahara Regions may not be an accurate criteria of comparison as similar limitations may have been encountered in those regions.

Hence, the authors should strongly recommend changing the policy of access to good quality ultrasound facility for all women in the first and second trimesters mainly and to emphasize the NGO's and governments to strongly implement upon the introduction of ultrasound in the first and second trimesters.

This will enable to get a more accurate data for the incidence of fetal and placenta pathologies in the concerned population.

7. PLOS authors have the option to publish the peer review history of their article (what does this mean?). If published, this will include your full peer review and any attached files.

Reviewer #1: No

Reviewer #2: No

---

## [Author Response · Author response to Decision Letter 1]

5 Aug 2024

Dear Editor,

We would like to thank you and the reviewers for their insightful and constructive feedback on our previous resubmission. In the enclosed document, we have provided detailed responses to all comments from the reviewers in this second round of revisions. The suggested revisions have been incorporated into the manuscript, with changes marked using track changes.

We look forward to your feedback and thank you for your consideration.

Best regards, 

Julius Emmrich

REVIEWER 1

Q1: Line 71 - 79: The paragraph included on foetal and placental abnormalities requires supporting references. Kindly include recent references to support the statements provided.

A1: We agree with the reviewer that the previous version of the paragraph lacked references. We have included relevant references into the paragraph as follows:

Obstetric ultrasound is a non-invasive, sensitive, accurate and cost-effective imaging technique for the detection and diagnosis of congenital abnormalities, obstetric pathologies, and complications of pregnancy. The use of obstetric ultrasound is proliferating in resource-constrained settings due to its increased affordability, availability, portability, and durability [1]. Identifying foetal and placental abnormalities during pregnancy is essential to identify potential complications during gestation or delivery [2]. Foetal abnormalities, e.g., congenital malformations can necessitate additional provisions for delivery care, for example by requiring a Caesarean Section [3]. Equally, placental abnormalities such as placenta praevia or placenta accreta may require additional precautions during delivery [2]. Placental abnormalities may affect the development of the foetus in utero, requiring additional antenatal or postnatal care and supervision [4]. As such, accurate information on placental and foetal abnormalities that can be obtained from obstetric ultrasounds is vital for adequate antenatal support, delivery planning and reducing maternal and neonatal mortality.

References:

1. Sippel S, Muruganandan K, Levine A, Shah S: Use of ultrasound in the developing world. Int J Emerg Med 2011, 4(72)

2. Goley SM, Sakula-Barry S, Adofo-Ansong N, et al. Investigating the use of ultrasonography for the antenatal diagnosis of structural congenital anomalies in low-income and middle-income countries: a systematic review. BMJ Paediatr Open. 2020;4(1):e000684. Published 2020 Aug 20. doi:10.1136/bmjpo-2020-000684

3. Wataganara, T., Grunebaum, A., Chervenak, F. & Wielgos, M: Delivery modes in case of fetal malformations. Journal of Perinatal Medicine 2017, 45(3): 273-279. https://doi.org/10.1515/jpm-2015-0364

4. Kim, E.T., Singh, K., Moran, A. et al. Obstetric ultrasound use in low and middle income countries: a narrative review. Reprod Health 15, 129 (2018). https://doi.org/10.1186/s12978-018-0571-y

Q2: Line 115 - 121: Suggestion to use past tense, the aim of this study was to.. as you are now reporting on the findings

A2: We thank the reviewer for this suggestion and have adapted the tense of the paragraph accordingly.

The aim of this study was to describe the demographic characteristics of a large group of pregnant women who utilized obstetric ultrasound services provided through an NGO. Specifically, we aimed to determine the prevalence of foetal and placental abnormalities in this study sample and to assess the timing and frequency of ultrasound examinations among these women. We expect these data to guide policymakers and program implementers in refining the design, implementation, and resource allocation of maternal health initiatives in this setting.

Q3: Line 141: Include a reference for Open Street Map as per the Journal guidelines

A3: We thank the reviewer for this suggestion and have changed the subtitle from “study sample” to “study cohort”

Q4: Line 248: Study sample can read study cohort

A4: We thank the reviewer for this suggestion and have changed the subtitle from “study sample” to “study cohort”

Q5:For the study sample section it will be easier to understand the numbers if depicted in a flow diagram. Kindly consider including one so that the readers can better understand the flow and final selection of study participants

A5: We thank the reviewer for the suggestion of including a flow diagram. We have uploaded such a diagram with our submission and added the following description of the diagram to the manuscript:

"Figure 2: Study cohort. Depiction of study sample of women included in the study between July 1, 2017, and September 30, 2020, in urban (Analamanga) and rural (Atsimo-Andrefana, Anosy, and Androy) Madagascar."

Q6: Line 283: there is a bracket close but not sure where the opening bracket is

A6: The opening bracket for the closing bracket after the end of the sentence can be found after “respectively”, highlighted in bold below:

Secondly, we estimated the proportion of four groups of foetal and placental abnormalities, defined as (i) abnormal placenta position, (ii) number of foetuses above one, (iii) amniotic fluid disorders, and (iv) other placental and foetal pathologies, respectively (e.g., placental adherence disorders, foetal malformations, and death).

Q7: Line 302 and 303: The correlations can be depicted as graphs as this is briefly mentioned in Line 398 - 401. It is a important finding; and later discussed in line 477-480.

A7: We thank the reviewer for recognising the importance of these findings. As most of the analyses conducted involved categorical variables, we perceived tables as a more adequate method of depicting these analyses. We have uploaded these tables as Supplementary Table 1 and added the following description to the manuscript: "Supplementary Table 1. Distribution of foetal and placental abnormalities and the number of ultrasound examinations in our study sample. The analyses revealed no significant correlation between the number of foetuses, amniotic fluid disorders, or placental and foetal pathologies and the number of ultrasound visits. Pearson’s chi-square test revealed a significant correlation between the grade of placenta praevia and the number of revisits (p <0.01)."

REVIEWER 2

Q1: The manuscript has been revised and queries have been answered, but the limitation of the manuscript is quite big. The results drawn are not representing the exact fetal and placental pathology of the population due to the various reasons described in the limitations the most highlighted being the timing of ultrasound which is mainly 3rd trimester. Similarity of results with other Sub Sahara Regions may not be an accurate criteria of comparison as similar limitations may have been encountered in those regions. Hence, the authors should strongly recommend changing the policy of access to good quality ultrasound facility for all women in the first and second trimesters mainly and to emphasize the NGO's and governments to strongly implement upon the introduction of ultrasound in the first and second trimesters. This will enable to get a more accurate data for the incidence of fetal and placenta pathologies in the concerned population.

A1: Thank you for your thoughtful feedback and for acknowledging the revisions made to our manuscript. We appreciate your concerns regarding the limitations of our study, particularly the timing of the ultrasound assessments predominantly being in the third trimester. 

We agree that the timing of ultrasound examinations can significantly influence the accuracy of detecting fetal and placental pathologies. Our study acknowledged this limitation and we understand that it may affect the generalizability of our findings. We have added an additional sentence regarding the comparability of our findings to lines 491ff. of the discussion section:

“We do acknowledge however, that this may not be an accurate comparison, as studies conducted in similarly resource-constrained settings in SSA may have been affect by similar limitations.”

However, as we describe in our manuscript, we believe our findings, describing obstetric pathologies in a severely understudied region, are of relevance for policy-makers and implementers alike.

As suggested by the reviewer, we have added a recommendation on accessing ultrasounds earlier in pregnancy to the conclusions section of our manuscript, which now reads:

“Our data show that obstetric ultrasound in a resource-restricted setting can lead to the detection and diagnosis of high-risk obstetric conditions including placental abnormalities, malpresentation of the foetus, and multiple gestations, which are highly associated with maternal and foetal morbidity and mortality. In stark contrast to current WHO recommendations, our findings suggest that universally accessible obstetric ultrasound is commonly and mostly utilised by pregnant women, especially those from rural, resource-constrained settings, during the third trimester of pregnancy. This finding highlights the need to improve policy and practice in Madagascar to improve access to obstetric ultrasounds, particularly during the first and second trimesters.”

---

## [Decision Letter · Decision Letter 2]

3 Sep 2024

PONE-D-24-05563R2Proportion of foetal and placental abnormalities in Madagascar: a cross-sectional study of 35,919 women at public-sector primary healthcare facilities in central and southern Madagascar, 2017-2020PLOS ONE

Dear Dr. Emmrich,

Thank you for submitting your manuscript to PLOS ONE. After careful consideration, we feel that it has merit but does not fully meet PLOS ONE’s publication criteria as it currently stands. Therefore, we invite you to submit a revised version of the manuscript that addresses the points raised during the review process.

**ACADEMIC EDITOR: Please respond to all reviewers comments**==============================

We look forward to receiving your revised manuscript.

Kind regards,

Ahmed Mohamed Maged, MD

Academic Editor

PLOS ONE

Reviewers' comments:

Reviewer's Responses to Questions

**Comments to the Author**

1. If the authors have adequately addressed your comments raised in a previous round of review and you feel that this manuscript is now acceptable for publication, you may indicate that here to bypass the “Comments to the Author” section, enter your conflict of interest statement in the “Confidential to Editor” section, and submit your "Accept" recommendation.

Reviewer #3: (No Response)

Reviewer #4: (No Response)

Reviewer #5: All comments have been addressed

Reviewer #6: (No Response)

2. Is the manuscript technically sound, and do the data support the conclusions?

Reviewer #3: No

Reviewer #4: Yes

Reviewer #5: Yes

Reviewer #6: Partly

3. Has the statistical analysis been performed appropriately and rigorously? 

Reviewer #3: N/A

Reviewer #4: Yes

Reviewer #5: Yes

Reviewer #6: Yes

4. Have the authors made all data underlying the findings in their manuscript fully available?

Reviewer #3: No

Reviewer #4: Yes

Reviewer #5: Yes

Reviewer #6: (No Response)

5. Is the manuscript presented in an intelligible fashion and written in standard English?

Reviewer #3: Yes

Reviewer #4: Yes

Reviewer #5: Yes

Reviewer #6: No

6. Review Comments to the Author

Reviewer #3: Due to the manuscript's localized focus, lack of originality, and insufficient contribution to the broader field, it is recommended for rejection. The study does not present new knowledge or significantly advance the existing body of literature, which is a critical criterion for publication in this journal

Reviewer #4: The world health recommendation of ultrasound scan before 24 weeks of gestation is to screen for congenital abnormalities, in the current manuscript late ultrasound scans were performed in preparation for delivery - the two uses of the ultrasound scan in pregnancy (screening for congenital anomalies vs preparation for delivery don't match).

Line 49-50: The proportion of malposition is reported and breakdown of those malapositions are presented as proportions of the total number of malpositions found. I would rather the authors present the constituent malpositions as proportions of the total number of malpositions found. Otherwise teh denominators get confusing for the reader.

In the methods section the authors should describe what happened to participants with more than one ultrasound scan - say early and late ultrasound scan - which of the two was analysed? How was teh selection for analysis arrived at?

In the results sections, authors report 492 participants has missing data - which type of data was this? Was it demographic characteristics, distance from the health facility, or reports missing critical information? It would be more informative for the reader if this missing data is defined.

In the discussion, line 328 - over 35,000 women; I would rather the authors re-state their total sample size.

Reviewer #5: Thank you for your understanding despite my delay in trying to go through the corrected work and initial reviewers suggestions, I found that the authors have made substantial improvement in the manuscript compared to the initial submission. Most of the corrections were done.

Reviewer #6: Dear Authors,

I am delighted to review your manuscript Titled "Proportion of foetal and placental abnormalities in Madagascar: a cross-sectional study of 35,919 women at public-sector primary healthcare facilities in central and southern Madagascar, 2017-2020" and find the research area to be highly interesting and relevant. However, to my knowledge, the manuscript is not well organized. Below, I have outlined my comments and suggestions for your consideration:

Major Comments:

1. Line 1 "Tittle)

from the scientific point of view, Placental abnormalities refer to any deviations from the normal structure or function of the placenta during pregnancy. These can be classified into several types:

Structural Anomalies: These include conditions like a succenturiate lobe (an extra lobe of the placenta) or velamentous cord insertion (where the umbilical cord attaches to the fetal membranes rather than the placenta itself).

Implantation Anomalies: These include placenta accreta (where the placenta attaches too deeply into the uterine wall) and placenta previa (where the placenta covers the cervix).

Functional Anomalies: These include placental insufficiency, where the placenta does not function properly, leading to inadequate supply of oxygen and nutrients to the fetus.

among these, you only focused on implantation Anomaly. so it is better if you modify your title and objectives accordingly.

2. Abstract Section

I noticed that the research question might not have been clearly addressed. Could you provide a single answer to the question: What is the proportion of fetal and placental abnormalities? Please ensure the conclusion of the abstract also reflects this information.

3. Line 62: Which keywords best describe your objectives? I couldn’t find keywords that reflect your title.

4. BACKGROUND: Lines 65-69: The paragraph does not support your research question. Your focus is on identifying fetal-placental abnormalities, not the importance of obstetric ultrasound. For instance, if there were other mechanisms or technologies to identify fetal-placental abnormalities, discussing ultrasound would be unnecessary. Additionally, there are unrelated sentences in the paragraph. Please rewrite the background section accordingly.

5. Method: Some mothers visited and had ultrasound scans more than twice. Does this make the study cross-sectional or cohort? What is your perspective?

6.What is the outcome variable in this context? How did you use the chi-square test? What are the independent variables, and how many were tested? These points were unclear to me.

Result:

7. Lines 299-300 seem more like a discussion or personal opinion.

Discussion

8. In this paper, I didn't capture clear outcome variable, which is why the authors discussed all the findings. Even it is too long to read and understand. Please rewrite the discussion again.

Conclusion:

9. the conclusion of these findings is unclear. It is necessary to determine whether the proportion is high, acceptable, or low to understand the implications. Lines 504-510 do not reflect the primary objectives of the authors.

10. ACKNOWLEDGEMENTS: Line 514: What does the author, Prof. Dr. Dr., mean? Is there a recognition of dual doctorates in Madagascar?

Authors have omitted the declaration section, which includes crucial components such as the source of funding and other pertinent information.

Minor

The terms “prevalence” and “proportion” are used interchangeably in the document. Please choose one and use it consistently.

In summary, the authors seemed confused and struggled to incorporate the importance of antenatal care, distance from health institutions, the significance and timing of ultrasound, and fetal and placental abnormalities. These topics were all presented together in the abstract, background, results, discussion, and conclusion sections, making them appear as outcome variables. This is outside the scope of the research question

Thank you

7. PLOS authors have the option to publish the peer review history of their article (what does this mean?). If published, this will include your full peer review and any attached files.

Reviewer #3: No

Reviewer #4: **Yes: **Musa Sekikubo, Senior Lecturer, Department of Obstetrics and Gynaecology, School of Medicine, Makerere University

Reviewer #5: No

Reviewer #6: No

---

## [Author Response · Author response to Decision Letter 2]

22 Sep 2024

Dear Editor,

We appreciate the reviewers' feedback on our previous resubmission. In the attached document, we have provided detailed responses to all of the reviewers' comments in this third round of revisions. We have incorporated the suggested changes into the manuscript, with all revisions clearly marked using track changes.

We look forward to your feedback and thank you for considering our manuscript.

Dr Julius Emmrich

 

Reviewer #3

Q1: Due to the manuscript's localized focus, lack of originality, and insufficient contribution to the broader field, it is recommended for rejection. The study does not present new knowledge or significantly advance the existing body of literature, which is a critical criterion for publication in this journal

R1: We respectfully disagree with the reviewer's assessment for the following reasons:

1. Large dataset: Our study analyses 35,919 pregnancies, making it the largest and most comprehensive obstetric ultrasound dataset from Madagascar and one of the few at this scale in Sub-Saharan Africa. It provides robust maternal health data in a severely resource-constrained setting. 

2. Global relevance: While focused on Madagascar, our findings on foetal malpresentation, placental abnormalities, and late obstetric screening have broad implications for similar settings in other low- and middle-income countries. In particular, our study exposes stark discrepancies between current WHO recommendations and real-world practices in LMICs.

3. Policy implications: By documenting the high prevalence of late-pregnancy ultrasounds in contrast to WHO recommendations for early screening, our study provides evidence-based observations for health policy reform. These findings are critical for improving maternal and neonatal care strategies in Madagascar and similar resource-limited settings globally.

4. Peer recognition: Five previous reviewers have recommended this manuscript for publication in PLOS One, acknowledging its scientific merit and contribution to the field.

Reviewer #4 

Q1: The world health recommendation of ultrasound scan before 24 weeks of gestation is to screen for congenital abnormalities, in the current manuscript late ultrasound scans were performed in preparation for delivery - the two uses of the ultrasound scan in pregnancy (screening for congenital anomalies vs preparation for delivery don't match) 

R1: We agree with the reviewer that there is a discrepancy between the recommended timing of ultrasounds by the WHO and the timing when most women in our study sample accessed ultrasound services.

As this is an observational study however that draws on existing data from an NGO intervention, there was no possibility to influence the timing of ultrasounds in the study sample. Despite mostly accessing ultrasound scans late in pregnancy, for most women in our sample these were still their first or only obstetric ultrasound examinations. Hence screening for congenital abnormalities where still conducted and reported even if the examination was conducted after 24 weeks gestation. 

Q2: Line 49-50: The proportion of malposition is reported and breakdown of those malapositions are presented as proportions of the total number of malpositions found. I would rather the authors present the constituent malpositions as proportions of the total number of malpositions found. Otherwise teh denominators get confusing for the reader. 

R2: We agree with the reviewer that the previous reporting of our findings was not clear and have amended the denominator accordingly. Lines 48ff now read:

“Foetal malpresentation at 36 weeks of gestation or later was diagnosed in 5.48% (176/3,211) of women with the breech presentation being most common (breech 72.73%, 128/176; transverse 15.34%, 27/176; mobile 9.01%, 16/176; oblique 2.84%, 5/176).”

Q3: In the methods section the authors should describe what happened to participants with more than one ultrasound scan - say early and late ultrasound scan - which of the two was analysed? How was teh selection for analysis arrived at? 

R3: We would like to draw the reviewer’s attention to lines 225 to 239 of the manuscript where we explain how we identified and matched the reports of study participants who received more than one ultrasound examination, as well as lines 268 to 283 of the manuscript where we detail our approach in analysing the data from women with multiple ultrasound visits. Briefly, all data from both visits were analysed in the paper, and the denominator was either the number of total patients enrolled in the study (when analysing ultrasound findings that do not change over time, e.g. the number of foetuses) of the total number of ultrasound examinations done, including re-visits (when analysing ultrasound findings that can change over time, e.g. foetal presentation. 

Q4: In the results sections, authors report 492 participants has missing data - which type of data was this? Was it demographic characteristics, distance from the health facility, or reports missing critical information? It would be more informative for the reader if this missing data is defined. 

R4: We would like to draw the reviewer’s attention to lines 247ff. of the results section, where we detail the missing data of our study participants:

“Of these 38,688 reports, 492 lacked data on the time of the first ultrasound, and 827 lacked data on placenta location. Of these 35,919 women, 762 had missing information on the number of foetuses. Of all foetuses examined, information on amniotic fluid volume was missing in 1,012 cases.”

Q5: In the discussion, line 328 - over 35,000 women; I would rather the authors re-state their total sample size. 

R5: We agree with the reviewer and have changed the “over 35,000 women” to “35,919 women” in line 391 of the manuscript.

Reviewer #5 

Thank you for your understanding despite my delay in trying to go through the corrected work and initial reviewers suggestions, I found that the authors have made substantial improvement in the manuscript compared to the initial submission. Most of the corrections were done. We thank the reviewer for their time and effort in providing us with valuable feedback on our work and are happy that they recognise the effort we have made in improving our manuscript according to the feedback received.

R: We thank the reviewer for their time and effort in providing us with valuable feedback on our work and are happy that they recognise the effort we have made in improving our manuscript according to the feedback received.

Reviewer #6 

Q1: 1. Line 1 "Tittle)

from the scientific point of view, Placental abnormalities refer to any deviations from the normal structure or function of the placenta during pregnancy. These can be classified into several types:

Structural Anomalies: These include conditions like a succenturiate lobe (an extra lobe of the placenta) or velamentous cord insertion (where the umbilical cord attaches to the fetal membranes rather than the placenta itself).

Implantation Anomalies: These include placenta accreta (where the placenta attaches too deeply into the uterine wall) and placenta previa (where the placenta covers the cervix).

Functional Anomalies: These include placental insufficiency, where the placenta does not function properly, leading to inadequate supply of oxygen and nutrients to the fetus. among these, you only focused on implantation Anomaly. so it is better if you modify your title and objectives accordingly. 

R1: We thank the reviewer for highlighting this important point and agree that we only focused our analysis on one dimension of placental abnormalities. We have therefore amended the title, short title, abstract and manuscript, to reflect “placental implantation abnormalities” instead of “placental abnormalities” more broadly.

Q2: I noticed that the research question might not have been clearly addressed. Could you provide a single answer to the question: What is the proportion of fetal and placental abnormalities? Please ensure the conclusion of the abstract also reflects this information. 

R2: We agree with the reviewer that the abstract in it’s current form lacked a concise answer to the research question. We have therefore added the following sentence to lines 52 to 53 of the abstract: 

“In a total of 1,284 cases (3.65%) a foetal of placental implantation abnormality was found.”

We have further amended the conclusions section of the abstract to reflect this statement:

“The proportion of foetal and placental implantation abnormalities detected by obstetric ultrasound was 3.65% of all cases in our study, aligning with findings from other countries in sub-Saharan Africa.”

Q3: Line 62: Which keywords best describe your objectives? I couldn’t find keywords that reflect your title. 

R3: We agree with the reviewer that our current selection of key words do not reflect our study’s title.

We have therefore added the following keywords to the list: 

“Madagascar; foetal abnormalities; placental implantation abnormalities”

Q4: Lines 65-69: The paragraph does not support your research question. Your focus is on identifying fetal-placental abnormalities, not the importance of obstetric ultrasound. For instance, if there were other mechanisms or technologies to identify fetal-placental abnormalities, discussing ultrasound would be unnecessary. sPlease rewrite the background section accordingly. 

R4: Thank you for your thoughtful comment. While we understand the point raised, we feel that the paragraph, though not directly addressing the research question, is important in providing context. It helps to underscore the relevance of our study, particularly by illustrating the growing use and significance of obstetric ultrasound in low-resource settings, also ensuring the comparability of our findings to other similar environments. For this reason, we would prefer to keep the paragraph as it is.

Q5: Method: Some mothers visited and had ultrasound scans more than twice. Does this make the study cross-sectional or cohort? What is your perspective? 

R5: In a cross-sectional study, data are collected at a single point in time, which aligns with the design of our study analyzing ultrasound screenings over a specific period (2017-2020).

Although some women had multiple ultrasound scans, this does not change the design of our study. The purpose of follow-up ultrasounds is typically to monitor or verify conditions detected in initial scans, but the overall structure of our study remains a snapshot of conditions during the study period. The fact that some participants had more than one ultrasound does not make the study a cohort study, which would require following individuals prospectively over time to observe outcomes.

Thus, despite the repeated ultrasounds for some mothers, the study remains cross-sectional because its main objective is to provide a prevalence estimate during a defined period. We did not focus on long-term outcomes or changes over time.

Q6: What is the outcome variable in this context? How did you use the chi-square test? What are the independent variables, and how many were tested? These points were unclear to me. 

R6: We thank the reviewer for highlighting that the previous description of our use of the Chi-square test lacked clarity.

We have added the following detail to lines 288ff of the manuscript:

“Specifically, the chi-square test was applied to categorical variables (for example, number of examinations received vs. presence of placenta previa) summarised as a contingency table to understand whether there was a relationship between the two categorical variables."

Q7: Lines 299-300 seem more like a discussion or personal opinion. 

R7: We agree with the reviewer that the sentence in question appears to be more adequate for a discussion section. We have therefore removed it from the results section. 

Q8: In this paper, I didn't capture clear outcome variable, which is why the authors discussed all the findings. Even it is too long to read and understand. Please rewrite the discussion again. 

R8: We agree with the reviewer that from the previous presentation, it was not clear what the main outcome of our study was, and which were subordinate findings. We have therefore restructured the discussion section to report and discuss on the findings relevant to our main research question, the proportion of foetal and placental implantation abnormalities, first. Accordingly, lines 391 to 403 now read as follows:

“Using data collected from over 35,919 women, we estimated the proportion of foetal and placental implantation abnormalities among pregnant women attending obstetric ultrasounds at public-sector healthcare facilities in Madagascar. Less than 10% of women in our sample were diagnosed with a potential complication during pregnancy. Foetal malposition was present in 5.48% (176/3,211) of cases and other abnormalities of placental implementation or the foetus occurred in a total of 3.65% (1,284/35,919) of all ultrasound examinations. 

Our study further revealed two subordinate but relevant findings. First, most women in our sample came from rural areas. Secondly, less than 10% of women in our sample were diagnosed with a potential complication during pregnancy. Foetal malposition, breech position and abnormal position of the placenta were the most common findings. Lastly, most obstetric ultrasounds were conducted in the third trimester of pregnancy, especially among women from rural backgrounds.”

Q9: the conclusion of these findings is unclear. It is necessary to determine whether the proportion is high, acceptable, or low to understand the implications. Lines 504-510 do not reflect the primary objectives of the authors. 

R9: We agree with the reviewer that the conclusion sections as it was did not reflect essential information on our primary research objective. We have revised the Conclusion section accordingly. It now reads as follows:

“Our data show that obstetric ultrasound in a resource-restricted setting can be used for the detection of high-risk obstetric conditions including placental implantation abnormalities, malpresentation of the foetus, and multiple gestations, which. In our sample, these conditions where present in less than 10% of cases and thus as common as in other countries in Sub-Saharan Africa. As a secondary finding, our study showed that accessible obstetric ultrasound is commonly and mostly utilised by pregnant women during the third trimester of pregnancy, especially those from rural, resource-constrained settings, during the third trimester of pregnancy. This is in n stark contrast to current WHO recommendations and highlights the need to ammend policy and practice in Madagascar to improve access to obstetric ultrasounds, particularly during the first and second trimesters.”

Q10: What does the author, Prof. Dr. Dr., mean? Is there a recognition of dual doctorates in Madagascar? 

R10: Prof. Dr. Dr. Bärnighausen holds two doctorate degrees, one in history of medicine from the University of Heidelberg, and one in Population and International Health from the

Harvard T.H. Chan School of Public Health at Harvard University.

Q11: Authors have omitted the declaration section, which includes crucial components such as the source of funding and other pertinent information. 

R11: As per PLOS One submission requirements, we have not included the declarations section within the manuscript but have uploaded the declarations separately in the submission system (https://journals.plos.org/plosone/s/submission-guidelines#loc-additional-information-requested-at-submission). However, we have included the key declarations here for the reviewer:

Ethics approval and consent to participate

Ethics approval was obtained from the Ethics Committee of the University of Heidelberg (registration number S-854/2020). All methods were carried out in accordance with relevant guidelines and regulations. Informed consent was waived by the Ethics Committee of the University of Heidelberg due to retrospective nature of the study.

Competing interests

The authors declare that

---

## [Editor Report · Decision Letter 3]

27 Sep 2024

Proportion of foetal and placental implantation abnormalities in Madagascar: a cross-sectional study of 35,919 women at public-sector primary healthcare facilities in central and southern Madagascar, 2017-2020

PONE-D-24-05563R3

Dear Dr. Emmrich,

We’re pleased to inform you that your manuscript has been judged scientifically suitable for publication and will be formally accepted for publication once it meets all outstanding technical requirements.

Kind regards,

Ahmed Mohamed Maged, MD

Academic Editor

PLOS ONE
---

## [Editor Report · Acceptance letter]

7 Oct 2024

PONE-D-24-05563R3 

PLOS ONE

Dear Dr. Emmrich, 

I'm pleased to inform you that your manuscript has been deemed suitable for publication in PLOS ONE. Congratulations! Your manuscript is now being handed over to our production team.

Kind regards, 

on behalf of

Professor Ahmed Mohamed Maged 

Academic Editor

PLOS ONE